# Numerical Investigation on Protective Mechanism of Metal Cover Plate for Alumina Armor against Impact of Fragment by FE-Converting-SPH Method

**DOI:** 10.3390/ma16093405

**Published:** 2023-04-27

**Authors:** Linlong Dou, Liling He, Yihui Yin

**Affiliations:** 1Institute of Systems Engineering, China Academy of Engineering Physics, Mianyang 621900, China; 2Department of Civil Engineering, Sichuan College of Architectural Technology, Deyang 618000, China; 3Shock and Vibration of Engineering Materials and Structures Key Laboratory of Sichuan Province, Mianyang 621999, China

**Keywords:** FE-converting-SPH, AD995, fragment, ballistic performance, cover plate thickness

## Abstract

It is of extreme importance to develop a reliable numerical prediction technique to simulate the ballistic response of ceramic armor subjected to high-velocity impact (HVI) to economize the test cost and shorten the design period. In the present manuscript, a series of experiments on tungsten heavy alloy (WHA) fragment’s penetration into 99.5% alumina (AD995) armors are systematically simulated by employing the FE-converting-SPH technique. The numerical results are compared with the experimental counterparts to find that the FE-converting-SPH method is fairly efficient in predicting the depth of penetration, the residual velocity, length and mass of fragment, and reproducing the crack forms of ceramic. The applicability and accuracy of the numerical model in terms of the algorithm, material model parameters and contact definitions are validated. Then, the relevant parameters of the calibrated numerical model are incorporated to explore the influence of cover-layer thickness on the armor performance. A few mechanisms regarding the cover plate have been identified to act on the armor performance, such as the alteration of fracture cone half-angle, proportion of energy absorbed by ceramic, mushrooming deformation of fragment, etc. The result of multi-mechanism superposition is that the best ballistic performance is endued with 1 mm cover-layer armor, which demonstrates a 24.6% improvement over the bi-layer armor with 4.96 g/cm^2^ area density, only at the cost of 15.7% increase in areal density, when back-plate thickness is held as 2 mm; for a constant area density of 4.96 g/cm^2^, a 1 mm cover-layer is also expected to be the best choice, with 10.7% improvement in armor performance.

## 1. Introduction

Both light weight and enhanced protective efficiency should be achieved for mobile equipment, such as combat vehicles, personal protective armors, aircrafts, etc. The high-velocity fragment generated by an explosion or other impact events with a high rate is one of the main threatening objects. The heavy tungsten alloy (WHA) fragment is one type of fragment used to study the anti-penetration performance of lightweight armor. In this scenario, the ceramic composite armor is a good choice for its low areal density and good anti-penetration performance.

The ceramic composite armor is commonly composed of ceramic face plate and metallic back plate. There are plenty of types of ceramics, i.e., alumina (Al_2_O_3_), carbon carbide (B_4_C), silicon carbide (SiC), etc. [1]. Alumina ceramic with 99.5% purity (AD995) is commonly adopted for its good performance, low cost and mature preparation technology. The back plate usually adopts ductile metallic or composite materials, e.g., aluminum alloy, steel, Kevlar, etc. Although there are a number of combinations of composite armors, the methodologies for studying the performances of composite armors are universal. In the present manuscript, we mainly concentrate on the performance of composite armors AD995/RHA (rolled homogeneous alloy).

Commonly, in composite armor, the ductile back plate can alleviate the negative influence of the intrinsic brittleness of ceramic. The high strength and high hardness of ceramic provide a good anti-penetration performance for the composite armor. For certain ceramic composite armor, there exists an optimal ballistic performance through the optimization of the configuration of armor [2], which needs a large number of research efforts.

Generally, the ballistic performance of the composite armor under high-velocity impact (HVI) is studied through theoretical analyses, experimental research and numerical simulations.

Experimentation is the most direct way to identify the ballistic performance of armor, which provides validation of theoretical and numerical models. The Nemat-Nasser team [3,4] have evaluated the ballistic performance of bare AD995 and AD995 with thin membrane restraint through recording the residual mass and residual velocity of the WHA fragment. Analogously, the anti-penetration performances of AD995/RHA armors against the WHA fragments with striking velocities ranging from 800~1250 m/s nominally are examined by comparing the residual mass, length and velocity of the fragment [5]. Depth of penetration (DOP) tests are conducted to investigate the effect of radial confinement and cover-layer on the ballistic performance of the armors subjected to the impact of the WHA fragment [6]. However, test results only provide the final features of the fragment or/and armor. High-speed photography or X-ray equipment should be adopted to record the penetration process; however, this has a high cost. Moreover, the experimental results are stochastic and untrusted for a smaller number of tests [7], which compels us to perform a series of experiments in order to obtain a reliable evaluation of performance and damage features. Additionally, the preparation of tests is commonly a long-term process. In a word, the experimental study is necessary but has a high cost and is time-consuming.

An efficient way is to predict the ballistic performance of armor through a theoretical model. However, each theoretical model is constructed based on a bunch of assumptions or premises. Moreover, certain unknown parameters in the theoretical model should be fitted through test results. For example, the Florence model [8] is constructed based on the total energy balance and the Zaera and Sanchez-Galvez model [9] is on the basis of Tate’s penetration equation. Both of them ignore the mushrooming of the penetrator and assume the constant half-angle of the ceramic fracture cone, for 63° and 65°, respectively. However, the cone half-angle, which depends on the velocity, shape and size of penetrator, and ceramic/backplate thickness, etc., is generally changeable between 20~75° [10,11]. Thereafter, although the mushrooming deformation of the penetrator and the variable half-angle are considered and incorporated in Zaera and Sanchez-Galvez model by Feli et al. [12], the determination of the half-angle needs to be obtained in specific experimental observation, which undoubtedly limits the wide application of the model. Afterwards, based on the method of energy balance, and integrating with many energy dissipation mechanisms, such as mushrooming deformation and the erosion of the projectile, compression and shear failure of ceramic, perforation of metal or the composite back plate, etc., the analytical prediction model of ceramic/backplate bi-layer armor is finally derived [13]. Nevertheless, the predictions of ballistic performance through the above models are valid just for small or/and medium-caliber penetrators and bi-layer ceramic armor without a cover-layer. Accordingly, the applicable range of the theoretical model is limited. Similar to experimental research, the majority of the theoretical models concentrate on the terminal performance of armor, such as DOP in armor, the residual velocity of the fragment, etc. The evolution of stress distribution and the failure pattern of armor, which is essential for understanding its protective mechanism, cannot be obtained.

Typically, the numerical simulation can compensate the evolution process of dynamic response of armor, which is usually absent in the experimental and theoretical studies. However, the accuracy of the simulation depends on the geometry and material models and algorithm for interaction and failure criterion of materials. The geometry should be in accord with the research object, fortunately, which can be easily achieved for the problem of fragment into ceramic composite armor.

The material model should embody the strain and strain rate effects that the material endures during a penetration event. For the fragment into ceramic composite armor, the strain rate may be up to 10^5^/s. Typically, the material models of the fragment, back plate and cover plate should represent proper behavior at large strain and a high strain rate of up to 10^5^/s or even larger. Moreover, the suitable failure criterion should also be chosen. Commonly, the Johnson-Cook model [14] can satisfy the above requirements. Since the material behavior of ceramic is tension-compression asymmetry, dependent on pressure and brittle under tension, the material models developed by Johnson and Holmquist are commonly used to represent the behavior of ceramic, i.e., JH1, JH2 and JHB [15]. From the moderate to high impact velocity range, the JH2 model is capable of properly reproducing the deformation and fracture patterns of ceramic observed in tests [15], which is also adopted in the present manuscript.

In terms of algorithms of interaction, there are generally finite elements (FE), smooth particle hydrodynamics (SPH) and the FE-converting-SPH method. Typically, for the penetration of high-velocity fragments into ceramic composite armor, three methods can be used to successfully obtain the ballistic performance of armor as long as the model is validated by experimental results, for example, Ref. [16]. However, they are good at different aspects and have their own shortcomings.

The FE method based on the Lagrangian-mesh has been developed to simulate a large deformation response of protective structures subjected to HVI loading. It is also proven to be able to provide a good estimation of the anti-penetration performance of metal armor [17] as well as ceramic armor [18,19,20] through the validations of experimental data. Despite the great success resulting from high computational efficiency and accuracy, the FE method still faces some difficulties in addressing the interface moving, large deformation and material fracture [21]. Especially for the large deformation under an HVI event, mesh elements may be distorted. The direct consequence is an unacceptable numerical error, sharp increase in solution time, and, even worse, the unexpected termination of calculation program. In this way, these distorted elements will be deleted by the element removal method [22]. However, the conservation of mass and energy of the computation system cannot be maintained, and the interaction force declines sharply in the vicinity of deleted elements. Hence, the FE simulation probably leads to an incorrect prediction in the structural response [16]. As depicted in Refs. [7,23,24,25], the FE method provides a poor prediction in terms of crack formations, particularly pulverized particles and larger fragments of ceramic subjected to HVI loading.

The SPH method is a typical meshless method, developed by Gingold et al. [26]. It circumvents the bottleneck of the discontinuity of the element removal to address the large deformation in the FE method and has been applied to simulate the complex response of ceramic during HVI loading. The SPH method has made a good prediction in the failure pattern of ceramic [15,27,28,29]. For example, Chakraborty et al. [27] carries out numerical research on ceramic/metal targets impacted by steel fragments at different velocities and draws a conclusion that the SPH method successfully reproduces diverse failure patterns such as crack interweaving, tensile spalling, fracture cone formation and subsequent fragmentation of ceramic. Similarly, the main features of the back-spalling process of AD995 against a Ø4.8 mm hard-steel sphere at 767 m/s are captured clearly when the SPH method is adopted [28]. The response behaviors of ceramic/metal armors under HVI loading are examined thoroughly through the SPH method to compare the properties of three constitutive models JH1, JH2 and JHB, and it is demonstrated that the SPH method can make a precise prediction in spalling, the formation of conoid, fragmentation and crack branching of ceramic [15]. Besides the failure pattern, the SPH method can also obtain reasonable ballistic performance of armors. For example, Hedayati and Vahedi [29] have numerically predicted the residual velocities of the projectiles penetrating SiC/6061-T651 bi-layer armor at velocities between 350 m/s and 866 m/s, and the predicted results are in good agreement with analytical results. The experiment in Refs. [3,4], i.e., WHA fragment into a 12.7 mm-thick AD995 tile, is also reproduced by a simulation based on the SPH method [30], in which the residual velocity and mass of the fragment are matched well with test measurements. Although the SPH method has been proven to be successful in forecasting the failure pattern of the ceramic and ballistic performance of armor, the tensile instability must be alleviated by adding artificial viscosity [28], which slightly decreases the accuracy of the simulation. Moreover, the SPH method has extremely low computational efficiency to significantly limit the scale of the simulation model.

The coupling method of FE-converting-SPH makes a fair compromise between calculation accuracy and efficiency. In this method, geometry is first discretized by finite elements. When transformation criteria are satisfied, such as failure criteria, the specified elements automatically transform into SPH particles. These particles, having inherited the attributes of parent finite elements, continue to participate in the subsequent computations.

The FE-converting-SPH method is adopted to simulate the response of ceramics and has obtained reasonable ballistic performance of armor and the failure pattern of ceramic with comparatively better computational efficiency. Goh et al. [31,32] have adopted this method to simulate the dynamic response of SiC/4340 steel armor against long-rod impact and investigate the effects of the cover plate and confinement from lateral pre-stress, and it turns out that numerically acquired DOP is a good match with test observations. Giglio et al. [33,34] have validated their FE-converting-SPH model by test results of 12.7 mm-thick AD995 tile impacted by WHA fragment at velocity 903.9 m/s in Refs. [3,4]. They have obtained the residual velocity of the fragment and fracture pattern of ceramic, which are consistent with test observations. More recently, bi-layer composite armors alumina/6061-T6Al [35] and alumina/Kevlar 29 [36] impacted by a 7.62 mm P80 bullet are simulated by the FE-converting-SPH method to acquire the residual velocities of the bullet, which match the test results well.

In a word, the FE-converting-SPH method is quite appropriate in simulating the complex failure phenomenon of ceramic and ballistic performance of armor. However, as evident from existing literatures, there is still very limited research [16,31,32,33,34,35,36] that focuses on using the FE-converting-SPH method to study the impact response of ceramic armor.

The main contribution and novelty of the present manuscript lies in that:(1)The FE-converting-SPH method will be adopted to simulate the dynamic response of AD995/RHA and AD995/4340 steel composite armors subjected to the impact of WHA fragment. The numerical model will be validated by various types of test results, focusing not only on the terminal ballistic performance, e.g., DOP in armor or the residual velocity of the fragment, but also on the fracture process and failure pattern of ceramic as well as the residual mass of the fragment.(2)Based on the numerical simulation, the optimization of the thickness of the RHA cover plate in AD995/RHA composite armor is subsequently carried out. Several affecting mechanisms involving cover-layer thickness are analyzed and identified, such as the pre-damage effect of the cover plate upon the ceramic, the shape change of fracture cone of ceramic, the prolonging of interaction time between fragment and ceramic, etc.(3)The numerical simulation helps to decrease the test number of verified tests and significantly shorten the time and cost of configuration optimization of armors.

## 2. Numerical Model

One numerical model is commonly comprised of a geometry model, material model, initial and boundary conditions, and interaction algorithms. For each problem, the geometry model and initial and boundary conditions should be in accord with the actual objects, which should be first depicted in each problem. For the material model, one principle is followed that material parameters are validated by ballistic experiments or obtained through reliable dynamic material tests. For the algorithm, as evinced in the above, the FE-converting-SPH method is adopted and the corresponding necessary controls are demonstrated.

### 2.1. Material Models

Five materials will be referred to in the present manuscript, i.e., AD995, WHA, 4340 steel, mild steel and RHA.

Firstly, for AD995 ceramic, the phenomenological JH2 model [37] is employed, which consists of the strength model, damage model and equation of state (EOS). The strength model defines the coupling relationships of the strength of intact and damaged materials. The corresponding parameters of JH2 model are listed in Table 1.

For the other four metallic types, i.e., WHA, 4340 steel, mild steel and RHA, the Johnson-Cook (JC) model is adopted. The JC model can represent the strain rate and thermal effects of material. Accompanied with JC fracture criterion and Mie-Grüneisen EOS, the fracture behavior of material can be described. Validated by impact tests and obtained by dynamic material tests, the corresponding parameters of the JC model are listed in Table 2 for WHA, 4340 steel, mild steel and RHA, respectively.

### 2.2. Interaction Algorithm Controls

In order to reproduce the failure patterns of ceramics, the FE-converting-SPH method is used to represent the dynamic response of ceramic. The other parts in the impact events, i.e., the cover plate, back plate, fragment, etc., are still represented by the FE method. This is similar to Refs. [31,32,33,34,35,36].

For the FE-converting-SPH method, one crucial point is to define the transformation criterion. In the present manuscript, the effective strain 0.06 is chosen to be the transformation criterion of ceramic, which is identical in Refs. [33,34]. The ceramic is first discretized by finite elements. Once the critical effective strain is achieved, the specified elements will transform into SPH particles. The transformation process will be accomplished by the keyword *DEFINE_ADAPTIVE_SOLID_TO SPH. Generally, one finite element is converted into one particle to avoid the substantial increase of solution time when converted into multiple particles [33,34]. The artificial viscosity coefficient Q1 = Q2 = 1 in keyword *HOURGLASS should be controlled for the newly generated particles in order to alleviate the tensile instability.

Contact definition is essential for the interaction between parts. The *CONTACT_ERODING_SURFACT_TO_SURFACE is adopted to control the contact between FE parts, and the *CONTACT_ERODING_NODES_TO_SURFACE is adopted to control the contact between SPH particles and the fragment. The *CONTACT_AUTOMATIC_NODES_TO_SURFACE is adopted to control the contact between particles and the cover/back plate. The bonding effect between ceramic and back plate and cover plate is controlled by *CONTACT_TIEBREAK_SURFACT_TO_SURFACE with tensile failure stress 120 MPa and shear failure stress 80 MPa, respectively [23]. 

At times, some distorted elements fail to be deleted by the JC model, which will induce the unexpected termination of computation. In order to maintain the stability of calculation, a large effective strain will be assigned to delete the distorted elements, e.g., 1.5 for RHA, 4340 steel and mild steel, 2.0 for WHA. The value is empirical and grid-dependent, which is decided by a series of calibration calculations.

### 2.3. Calibration and Validation of Numerical Models

Three types of impact tests are employed to calibrate the numerical model, as listed in Table 3. All tests have a normal impact gesture, and the impact point of fragment is on the middle point of the impact face of the ceramic. The fragment is fired by a gas gun in Ref. [4], smoothbore gun in Ref. [5] and powder gun in Ref. [6], respectively. In this way, the fragment has no rotation about its axis. The velocity when the fragment just hits the armor is defined as the initial velocity of fragment.

Ref. [4] describe that the armor has a free boundary condition, but Refs. [5,6] do not specify the boundary conditions of targets. Ref. [40] shows that when the ratio of the lateral dimension of ceramic to the diameter of the fragment is greater than 7, the influence of boundary conditions on the ballistic properties of plates can be ignored. Since the lateral dimension of ceramic in Refs. [5,6] is large enough, the free boundary condition is adopted for Refs. [4,5,6] in numerical modeling. In addition, an 8-node hexahedron element with one Gaussion integral point is used to make discrete the target and fragment for all numerical models. The three-dimensional models of ballistic experiments in Refs. [4,5,6] are presented in Figure 1.

In order to reproduce the dynamic response of bare AD995 tile in Refs. [3,4], we construct a 3-D full-scale model with a uniform mesh size 0.5 mm, which can avert the artificial path deflection of crack arising from irregular mesh as much as possible [34]. The fragment impactor also adopts a uniform mesh size of 0.5 mm, which is used as all subsequent mesh divisions of the impactor in this manuscript. The simulation results are listed in Table 4. They are in good accordance with test results with errors less than 6%. The morphologies of ceramic during penetration at typical times are listed in Figure 2, which are recorded by X-ray photography in Ref. [4]. It is evident that the front-face ejecta and back-face deformation of ceramic obtained from the simulation agree well with the experimental results. The damage morphology and crack distribution in AD995 tile are shown in Figure 3. The number of cracks on the front and back face are approximately 20, which is consistent with the test results in Refs. [3,4]. There is a conoid near the back face of ceramic, whose diameter is about 58 mm, which agrees well with the test result, i.e., 58.8 mm [3,4].

**Figure 2 materials-16-03405-f002:**
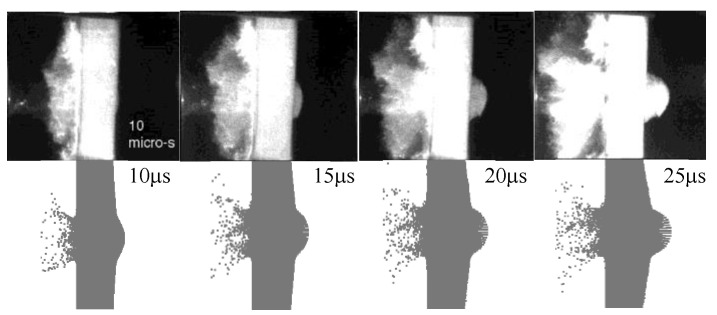
Comparisons of the X-ray images in Ref. [4] and numerical images of WHA fragment penetrating into a bare AD995 tile. **Top**: test images [4]; **bottom**: simulation images.

**Figure 3 materials-16-03405-f003:**
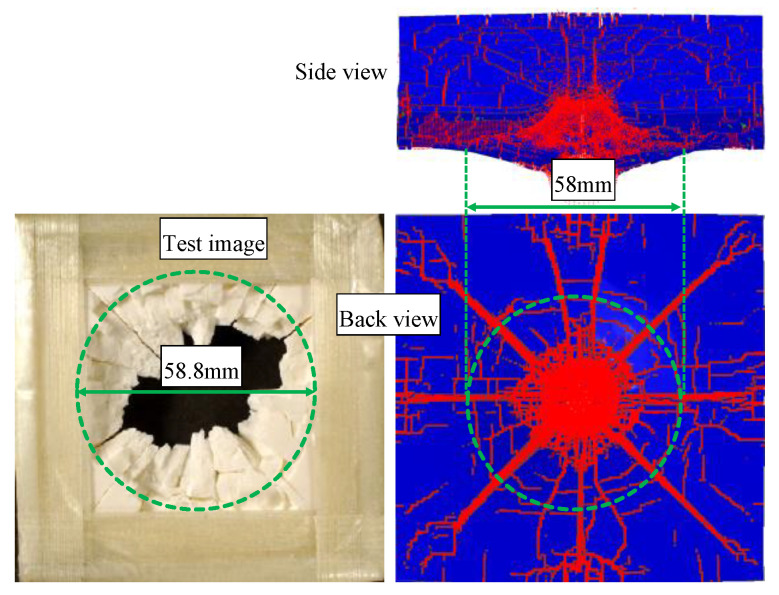
Damage morphology of bare AD995 tile after penetration. The area surrounded by the green circle indicates the crater. **Left**: test image [4]; **right**: simulation image.Moreover, the ballistic performances of composite armors AD995/RHA and (mild steel)/AD995/4340 in Refs. [5,6] are also simulated. The mesh size of the interaction zone in armor, i.e., three times the diameter of the fragment, is 0.5 mm. It gradually becomes rough from 0.5 mm in the center area to 2 mm at the boundary. The simulation results are listed in Table 5 and Table 6. It is clear that the numerical results are in good agreement with the test results.

**Table 5 materials-16-03405-t005:** Comparisons of test data [5] and numerical results.

Test No.	AD995/RHA Thickness (mm)	Initial Velocity of Projectile (m/s)	Residual Velocity of Projectile (m/s)		Error (%)	Residual Length of Projectile (mm)		Error (%)
			Test	Simulation		Test	Simulation	
1	10/15.7	794	432	432	0	36	32.2	10.6
2	10/15.7	1237	1028	1050	2.1	37	37.3	0.8
3	10/21	1245	967	984	1.8	34	32.1	5.6

**Table 6 materials-16-03405-t006:** The comparisons of experimental data [6] and numerical results for AD995/RHA composite armors with or without mild steel cover plate.

Test No.	Boundary Condition	Initial Velocity of Fragment (m/s)	DOP (mm)		Error (%)
			Test	Simulation	
1	Unconfined	1480	69.4	71.2	2.6
2	Unconfined	1500	72.3	71.8	0.7
3	Confined by 4340 steel surround	1520	69.0	69.4	0.6
4	Confined by 4340 steel surround	1550	69.0	69.4	0.6
5	Confined by 4340 steel surround with single mild steel cover plate	1530	65.8	63.7	3.2
6	Confined by 4340 steel surround with double mild steel cover plates	1470	65.8	65.6	0.3

In this way, the numerical model we adopt in the present manuscript is calibrated and validated by various tests. It obtains reliable ballistic performances of ceramic composite armors and the failure patterns of ceramic. The reliability of the numerical model is confirmed.

## 3. Configuration Optimization of AD995/RHA Composite Armor with RHA Cover Plate

A cover plate is usually placed in front of ceramic in order to enhance the protective efficiency of bi-layer ceramic composite armor. Generally, the metallic cover plate can delay and attenuate the impact stress acting on the ceramic [31,41], restrict the flow of the pulverized ceramic particles to increase the abrasion effect of the fragment [42] and prolong the dwell duration time [43]. However, there exists an optimal shielding effect for certain configuration of tri-layer composite armor [6,31], i.e., cover-plate/ceramic-plate/back-plate. Although the shielding mechanism of the cover plate has already been revealed through limited numerical simulations [44], the variation of the shielding effect of the cover plate with the configuration of tri-layer armor should be further studied.

In the present manuscript, we focus on the composite armor AD995/RHA with or without an RHA cover plate. The cross section of the armor is square with a side length of 101.6 mm. The threatening object is a cylindrical fragment made of WHA (6.14 mm × 20.86 mm). Two groups of armor are designed. One has constant areal density 4.96 g/cm^2^ for the tri-layer armor. The other is to cover the bi-layer AD995/RHA armor with RHA plates of different thicknesses for a 4.96 g/cm^2^ areal density of armor and 2 mm constant thickness back plate. The 12.7 mm-thick AD995 tile normally impacted by the fragment with velocity 903.9 m/s is chosen as the reference case, whose test result was reported in Refs. [3,4]. The numerical model calibrated in Section 2 is adopted to reveal the protective mechanism of the designed armors. Herein, all components in the numerical model are divided into a uniform mesh size of 0.5 mm.

### 3.1. Armors with Constant Areal Density 4.96 g/cm^2^

In order to obtain an optimal ballistic performance of bi-layer composite armor, the configuration of the armors should theoretically satisfy the Heterington Equation [2], i.e.,
(1)$$\left(\frac{8A-\rho_{ceramic}D}{2A+\rho_{ceramic}D}\right)\frac{\rho_{back}}{\rho_{ceramic}}=\frac{T_{ceramic}}{T_{back}}$$

Here A is the areal density of the bi-layer armor. $\rho_{back}$ and $\rho_{ceramic}$ are, respectively, the densities of the back plate and ceramic tile. D is the initial diameter of the fragment. $T_{back}$ and $T_{ceramic}$ are, respectively, the thicknesses of the back plate and ceramic tile. The areal density A is represented as
(2)$$A=\rho_{ceramic}T_{ceramic}+\rho_{back}T_{back}$$

Based on Equations (1) and (2), the bi-layer composite armor with an optimal performance can be designed. For the AD995/RHA bi-layer armor, in order to match the thickness of ceramic, the thickness of the back plate can be obtained based on Equations (1) and (2). Moreover, in design of the finite-thickness lightweight armor to protect against small caliber threats, the thickness of ceramic tile generally falls in 6 mm and 10 mm [36,45,46], and the areal density of the bi-layer armor should be less than or equal to 4.96 g/cm^2^. For the bi-layer armor with areal density less than 4.96 g/cm^2^, one RHA cover plate is added to form a tri-layer composite armor and compensate the areal density of the armor to achieve 4.96 g/cm^2^. The configurations of the armors we design are listed in Table 7. In order to form a fracture cone in ceramic to spread the force onto a relatively wide area of ductile back plate, the ratio of the thickness of ceramic to the diameter of the fragment should be as large as possible [47]. Moreover, the shielding effect of the cover plate is our main focus. In this way, we finally choose Type VI, i.e., cover/ceramic/back = 0.5/8.7/1.5.

In the following analyses, the thickness of ceramic is fixed as 8.7 mm, except the reference case. For simplicity, the configuration of bi- or tri-layer composite armor is represented by the thickness ratio of the cover and back plates, e.g., 0.5/1.5 for Type VI. Hereinafter, the same rule is followed unless otherwise stated.

For the same areal density of 4.96 g/cm^2^, we design five composite armors, as listed in Table 8. The total thickness of back and cover plates maintains 2 mm. Through numerical simulations, we obtain the ballistic performance of the armors, as listed in Table 8. The residual velocity and mass of fragments are recorded. Clearly, the fragment perforates all of the armor. Since the fragment has large mass loss, the residual velocity is not enough to identify the ballistic performance of armor. We choose the energy loss of the fragment as the identifier. The higher the energy loss of the fragment, the better the ballistic performance of the armor.

The energy loss of the fragment $E_{\mathrm{loss}}$ is defined as
(3)$$E_{loss}=\left[\frac{E_{k\_\mathrm{frag}\_0}-\left(E_{k\_\mathrm{frag}\_r}+E_{\mathrm{in}\_\mathrm{frag}\_r}\right)}{E_{k\_\mathrm{frag}\_0}}\right]\times 100\%=\left[1-\left(\frac{M_{r}V_{r}^{2}}{M_{0}V_{0}^{2}}-\frac{2E_{\mathrm{in}\_\mathrm{frag}\_r}}{M_{0}V_{0}^{2}}\right)\right]\times 100\%$$

Here $E_{k\_\mathrm{frag}\_0}$ and $E_{k\_\mathrm{frag}\_r}$ are, respectively, the initial and residual kinetic energy of the fragment. $E_{\mathrm{in}\_\mathrm{frag}\_r}$ is the internal energy of the residual fragment after perforating the armor. $M_{0}$ and $M_{r}$ are the initial and residual mass of the fragment, respectively. $V_{0}$ and $V_{r}$ are, respectively, the initial and residual velocity of the fragment. The simulation results of $E_{\mathrm{loss}}$ for fragments after perforating the armors are also listed in Table 8.

Clearly, 2.0/0.0 armor has the worst ballistic performance, which is even worse than the reference case, i.e., a bare ceramic tile. When the thickness of the cover plate varies from 0 mm to 1.5 mm, the ballistic performance of the composite armor with constant areal density 4.96 g/cm^2^ is better than the reference case. The 1.0/1.0 armor has the best ballistic performance, which is, respectively, improved by 15.3% and 10.7%, compared with the reference case and 0.2/2.0 bi-layer armor.

Further study reveals the temporal variation of energies of the fragment during perforation, as shown in Figure 4. The energy ratio is defined as the energy compared to the initial kinetic energy of the fragment. The energy ratio of the internal energy of fragment is less than 8% and has a rapid change in the first 35 μs, which indicates the large, localized deformation of the fragment. Thereafter, the internal energy of the fragment changes slowly and declines to less than 5%, which implies a small deformation of the fragment. For kinetic energy, its energy ratio sharply descends in the first 35 μs. After 35 μs, it descends very slowly. For the armors with a cover plate between 0.0 mm and 1.0 mm, the curves of the energy ratio for the kinetic energy of the fragment almost coincide for the first 21 μs. Thereafter they deviated from each other. Clearly, the total energy of the fragment resembles the kinetic energy of the fragment, for the internal energy is comparatively small. The fragment has the least total energy after perforating the armor 1.0/1.0.

It is assumed that the energy loss of the fragment is totally absorbed by the armor, i.e.,
(4)$$E_{k\_frag\_0}-\left(E_{k\_frag}+E_{in\_frag}\right)=({E_{k\_cov}+E}_{in\_cov})+\left(E_{k\_cer}+E_{in\_cer}\right)+\left(E_{k\_b}+E_{in\_b}\right)$$

Here $E_{k}$ and $E_{in}$ are the kinetic and internal energies, respectively. The subscripts frag, cov, cer and b represent the fragment, cover plate, ceramic and back plate, respectively.

The distributions of energies in the components of armor are shown in Figure 5. Apparently, the internal and kinetic energies of the cover plate increase with its thickness increasing. In the first 10 μs, the kinetic and internal energies of the cover plate reach their maximum. Thereafter, the kinetic energy descends to almost zero, and the internal energy remains almost constant. The plateau value of the internal energy, which is less than 7%, increases with the thickness of the cover plate increasing. For the back plate, the energy absorbed decreases with its thickness decreasing. However, the descending range between 2.0 mm and 1.0 mm is comparatively small. The energy absorbed by the cover and back plate gradually increases from 0.0/2.0, 0.5/1.5 to 1.0/1.0, as shown in Figure 5d. However, their differences are comparatively small. The armor 2.0/0.0 has the least absorbed energy of cover and back plates. The energy absorbed by the ceramic is higher than that absorbed by the cover and back plates for either bi-layer armor or tri-layer armor. It firstly increases and then decreases with the thickness of the cover plate increasing. Its maximum occurs for 1.0/1.0.

In this way, the energy distribution in armor changes with the configuration varying. The best ballistic performance appears with configuration 1.0/1.0, which improves the energy absorbed by the cover plate and back plate, as well as the ceramic.

Generally, enhanced ballistic performance of the composite armor demands small damage caused by the pressure of the cover plate, a large contact area between the fragment and the ceramic as well as a large fracture cone in the ceramic and enough of a constraint effect of the cover and back plates, which will increase the mass loss of the fragment and then boost the energy loss. In order to reveal the interaction mechanism between various armor components as well as the fragment, the deformation processes of armor and fragment at typical time points are shown in Figure 6.

Firstly, the fragment impacts the armor. The interaction time between cover plate and fragment is less than 5 μs, which increases with the thickness of the cover plate increasing. Clearly, the energy loss of the fragment in the first 5 μs is almost the same for four types of armors with cover plates, which is slightly smaller than the bi-layer armor without a cover plate, i.e., 0.0/2.0, as shown in Figure 4c. The head of the fragment mushrooms after perforating the cover plate. The diameter of the contact area between fragment and ceramic is larger than that of the original fragment for 0.5/1.5 and 1.0/1.0, as shown in Figure 6. Clearly, the thin cover plate will help increase the contact area between the fragment and ceramic for tri-layer armor. Moreover, the damage area in the ceramic caused by the pressure of the cover plate increases with the thickness of the cover plate increasing for the armor with the cover plate. The energy absorbed by the ceramic first increases and then remains almost unchanged in the first 5 μs for the tri-layer armor, as shown in Figure 5c. The bi-layer armor 0.0/2.0 has the best ballistic performance at 5 μs, as shown in Figure 4c.

At 15 μs, a fracture cone is formed in the ceramic. Apparently, the cone half-angle decreases with the thickness of cover plate increasing for the armor with cover plate. The maximum half-angle of the cone occurs for 0.5/1.5, i.e., 48.8°. The diameter of the top surface of the cone is also listed in Figure 6. Clearly, the diameter decreases with the thickness of the cover plate increasing. Furthermore, the cover plate will provide constraints for the fractured ceramic, refraining the fractured ceramic from ejecting out of the impact surface. As shown in Figure 6, the ceramic near the impact surface of the armor almost fractures into pieces for the bi-layer armor 0.0/2.0. There is severe damage for the ceramic in armor 0.5/1.5, too, which indicates the lack of confinement of the thin cover plate with thickness 0.5 mm. When the thickness increases to 1 mm for the cover plate, the ceramic near the impact surface has little damage, which is similar for thickness 1.5 mm. It indicates the good confinement of the cover plate. However, for 2.0/0.0, even though the cover plate confines the ceramic, there is no back plate to provide support or constraint over the ceramic. Therefore, the damage in ceramic propagates from the back of the ceramic to its impact surface, which is devastating for the ballistic performance of armor. Moreover, the armors 0.5/1.5 and 1.5/0.5, respectively, have the maximum and minimum mass loss of the fragment at 15 μs, as shown in Figure 7. In a word, several factors compete with each other during interaction between fragment and armor. Finally, the armor 0.5/1.5 exhibits the best ballistic performance at 15 μs. The armors 0.0/2.0 and 1.0/1.0 have good ballistic performances, which are better than the armors 1.5/0.5 and 2.0/0.0.

At 21 μs, the fragment penetrates into the fractured cone for armor 0.0/2.0. The mass loss of fragment for the armor 0.0/2.0 sharply increases, as shown in Figure 7, which is caused by the abrasion of the fracture cone. The head of the fragment becomes ogival. However, the fragment pushes the integral cone forward for the other types of armors, i.e., 0.5/1.5, 1.0/1.0, 1.5/0.5 and 2.0/0.0. In these armors, the head of the fragment is blunt, as shown in Figure 6. In this phase, the ceramic dominates the absorbed energy of armor, as shown in Figure 5c. Apparently, the armor 0.5/1.5 has the best ballistic performance at 21 μs. Similar performances are obtained for the armors 1.0/1.0 and 0.0/2.0.

At 35 μs, the fracture cone is pulverized for all armors studied. The fragment penetrates into the cone, and also is abraded by the cone. Apparently, the fragment penetrating the armor 1.0/1.0 has the least residual length and a blunted head. At this moment, it has the maximum mass loss of the fragment, as shown in Figure 7. For all other armors, the head of the fragment becomes ogival. The total energy of the fragment remains almost unchanged for the armor 2.0/0.0, as shown in Figure 4c; the unconfined and pulverized ceramic can provide little resistance force for the fragment. The back plates of the armors 0.0/2.0, 0.5/1.5, 1.0/1.0 and 1.5/0.5 bulge with high height, which indicates the ascending of the absorbed energy for the back plate, as shown in Figure 4b. However, ceramic still dominates the energy absorption of the armor, as shown in Figure 4c, which has already reached its maximum at this moment. For all armors studied, the energy loss of the fragment almost achieves its maximum at this moment. Only a little ascending of energy loss of the fragment appears after 35 μs, as shown in Figure 4c. The increase of energy absorption for the armor 1.0/1.0 may be caused by the confinement of the cover and back plates. The 0.5 mm-thick cover or back plate, i.e., armors 0.5/1.5 and 1.5/0.5, will not provide enough constraints for the fractured ceramic. Hence, the armor 1.0/1.0 obtains the best ballistic performance at 35 μs.

After 35 μs, the fragment continues to move forward. It breaks the back plate at 38 μs and 46 μs for armor 0.0/2.0 and 0.5/1.5, respectively. However, the back plate is broken by the pulverized ceramic for the armors 1.0/1.0 and 1.5/0.5. Finally, at 65 μs, the fragment has perforated each armor studied. Clearly, the back plate has petal flowering for the armor 0.0/2.0, 0.5/1.5 and 1.5/0.5, which indicates the failure mechanism as ductile hole enlarging. However, a plug is formed for the back plate in the armor 1.0/1.0, which is a representative failure pattern for ductile target impacted by a blunt projectile. The fragment has the least residual length and mass after perforating the armor 1.0/1.0. 

In a word, ceramic dominates the energy absorption of the armor, while the cover and back plate will change the performance of ceramic. In the earlier stage of penetration, the thin cover plate with thickness 0.5 mm generates a fracture cone with a large half-angle and comparatively large diameter of its top surface, which enhances the performance of ceramic. In the final stage of penetration, the confinement of ceramic provided by the cover and back plate plays the key role in the performance of the armor; it is the reason that renders 1.0/1.0 armor able to finally achieve the best performance. Therefore, it can be reasonably inferred that for armors with a constant areal density of 4.96 g/cm^2^, the restraint effect of the cover plate may be dominant in the ballistic resistance of armor on the premise of having a certain thickness of the back plate. Shockey et al. [10] similarly points out that the protective performance of armor is largely determined by the flow characteristics of the pulverized ceramic particles. It is advised that the thickness of the cover plate should be 1.0 mm for the tri-layer armor with an areal density of 4.96 g/cm^2^.

### 3.2. Armors with 2 mm-Thick Back Plate

The armor with a constant areal density of 4.96 g/cm^2^ changes the thickness of the back plate, which brings more than one variant and makes the mechanism of the cover plate more complicated. In this way, we add the cover plate on the bi-layer armor AD995/RHA with an areal density of 4.96 g/cm^2^, i.e., the thickness of the back plate is constantly 2 mm, as listed in Table 9. The ballistic performances are obtained by numerical simulation, which are also listed in Table 9.

Clearly, the performance of armor does not increase monotonously with its areal density increasing. The best performance appears in the armor 1.0/2.0 and increases by 34.7% and 24.6%, respectively, compared to the reference case and 0.0/2.0 bi-layer armor without a cover-layer. Similar to Section 3.1, the temporal curves of energy ratios for the fragment are shown in Figure 8. The final internal energy of fragment is less than 4%, which is much smaller than its kinetic energy. Correspondingly, the fragment after perforating the armor 1.0/2.0 has the least total energy.

The energy distribution in the components of the armor is shown in Figure 9. The variation of the energy absorbed by the cover plate is similar to the cover plate in the armor with a constant areal density of 4.96 g/cm^2^. It increases monotonously with its thickness increasing. The same trend appears for the energy finally absorbed by the back plate. However, for ceramic, its maximum energy absorption occurs for the armor 1.0/2.0. It indicates that the cover plate changes the performance of ceramic. The energy absorption of ceramic dominates the performance of the composite armor.

The interaction process between the fragment and armor is shown in Figure 10. Similar to Section 3.1, 0.0/2.0 armor has the best ballistic performance at 5 μs. The time when the fragment and ceramic contact is similar to the armor with the same cover plate is in Section 3.1. The damage caused by the pressure of the cover plate increases with the thickness of cover plate increasing. However, the armor with the same cover plate in Section 3.1 has more severe damage in ceramic, for the weaker support of the thinner back plate, as shown in Figure 6 and Figure 10.

At 15 μs, the fracture cone is formed in ceramic. The diameter of the top surface of the cone is 1.43D for all tri-layer armors, which is slightly smaller than that of the bi-layer armor 0.0/2.0, i.e., 1.67D. Compared to Figure 6, the thicker back plate will increase the diameter of the top surface of the cone. Moreover, the maximum half-angle of the cone appears for the armors 0.5/2.0 and 1.0/2.0, i.e., 48.5°, which is slightly larger than the armor 0.0/2.0. The armor 2.0/2.0 has the minimum half-angle, i.e., 35.5°, which may be caused by the pre-damage of ceramic. At this moment, 0.5/2.0 armor has the best performance, as shown in Figure 8c. Its ballistic performance is dominated by the performance of ceramic, as shown in Figure 9c.

At 25 μs, part of the fracture cone in ceramic is pulverized. The fragment penetrates into the cone for the armor 0.0/2.0. For the lack of confinement of the cover plate, the ceramic near the impact surface of ceramic fractures into pieces in the armor 0.0/2.0. The pulverized ceramic ejects out of the impact surface. For the armor 0.5/2.0, the fragment slightly penetrates into the fracture cone in ceramic. However, for the other armors, the fragment only pushes the cone forward with little penetration. At this moment, the armor 1.0/2.0 has the best ballistic performance, as shown in Figure 8c. The confinement of the cover plate enhances the performance of armor. Furthermore, for the armors 1.5/2.0 and 2.0/2.0, even though the cover plate provides enough constraints, the small volume of the fracture cone in ceramic devastates its ballistic performance.

Different from the armors in Section 3.1, the fragment finally penetrates into the fracture cone of ceramic and contacts with the back plate, which indicates the sufficient support provided by the 2 mm back plate. The contact time of the fragment and back plate increases from the armor 0.0/2.0 to 0.5/2.0 and 1.0/2.0. However, the time remains almost unchanged for the armors 1.0/2.0, 1.5/2.0 and 2.0/2.0, which indicates a slight decrease in the performance of ceramic with the thickness of cover plate increasing. It can also be found by calculation when the cover-layer thickness increases from 0 mm to 2 mm, the interaction time between the fragment and ceramic increases at first and then decreases and reaches the maximum at 1 mm. From this point of view, the armor with a 1 mm cover-layer deserves to provide more mass loss of fragment and better ballistic performance of armor. Furthermore, although the thick cover plate provides good confinement of ceramic, the small half-angle of cone limits the performance of ceramic.

The mass loss of the fragment is shown in Figure 11. Clearly, the fragment has the maximum mass loss in the earlier stage of penetration of armor 0.5/2.0. However, the mass loss rapidly achieves its maximum for the armor 0.5/2.0. According to Figure 10, the pulverized ceramic has relatively low abrasion for the armors 0.0/2.0 and 0.5/2.0, for its lack of axial confinement provided by the cover plate. However, for the armor with a cover plate thicker than 0.5 mm, the abrasive effect on the fragment is significantly enhanced by the confined pulverized ceramic, which causes 1.0/2.0 armor to accomplish the maximum mass loss of fragment.

In a word, the cover plate provides axial confinement of the ceramic. The thicker the cover plate, the higher the confinement of the ceramic. However, the thick cover plate is likely to damage the ceramic, which decreases the cone half-angle and then devastates the ballistic performance of armor. The best performance of the tri-layer composite armor is obtained by 1.0/2.0 armor.

Although it is difficult for us to provide a conclusive relation between the cover plate thickness and ballistic performance for ceramic composite armor, the cover plate thickness has been identified by our numerical implement to play a considerable role in the anti-penetration performance of ceramic armor. In order to improve the ballistic performance of the ceramic armor, the thickness of the cover plate is suggested as 1 mm. As depicted in Ref. [48], the thickness of the cover plate is suggested not to exceed 1 mm, which is consistent with our analyses.

## 4. Conclusions

It is urgent to develop an accurate numerical method to study the ballistic performance of ceramic composite armor, in order to reduce the high cost of the ballistic experiments. Since the FE-converting-SPH method achieves a balance of time-cost and accuracy, it is adopted in the present manuscript to simulate the dynamic response of ceramic composite armor subjected to HVI loading of the fragment. The numerical model is validated by the tests reported in open references. The shielding mechanism of the cover plate for the ceramic composite armor is investigated based on the validated numerical model. An optimal thickness of the RHA cover plate is obtained. The main conclusions are as follows:(1)Based on the FE-converting-SPH method, a specified numerical model for (RHA or mild steel)/AD995/RHA (or 4340 steel) against high-speed WHA fragment is constructed. The numerical model is validated by test results, including DOP in armor, the residual velocity of fragment, the failure pattern of ceramic as well as the residual mass of fragment. The deviation between test and simulation is less than 11%. Especially, with regard to the residual velocity and DOP, the numerical evaluation errors are all within 3.2%.(2)The shielding mechanism of cover plate is numerically investigated. Since the energy is mainly absorbed by the ceramic component, the shielding effect of cover plate is intrinsically to alter the ballistic performance of ceramic. It is indicated that the metallic cover plate will pre-damage the ceramic, which weakens the ballistic performance of armor. Moreover, the half-angle of fracture cone in ceramic will be changed with thickness of cover plate varying. The maximum is 48.8° for the armor 0.5/1.5. Correspondingly, the value is 48.5° for the armor 0.5/2.0 and 1.0/2.0. The increase of the half-angle of the fracture cone enhances the ballistic performance of ceramic. The cover and back plates provide the confinement of the fractured ceramic particles to increase the penetration resistance of fragment and prolong the interaction time between the fragment and ceramic. The 1.0/1.0 armor has the best confinement among the armors with an areal density of 4.96 g/cm^2^, whereas, 1.0/2.0 armor unfolds the best confinement for a constant 2 mm-thick back plate. The pre-damage effect, variation of half-angle of fracture cone and confinement of cover and back plates compete with each other.(3)For the composite armor with constant areal density 4.96 g/cm^2^, the best ballistic performance can be obtained with 1.0 mm cover-layer. It possesses an increment of 10.7% energy loss compared to 4.96 g/cm^2^-AD995/RHA armor. For the constant 2 mm-thick back plate, the best ballistic performance can also be attained by the 1.0 mm cover-layer. It has an increment of 24.6% energy loss compared to the 0.0/2.0 bi-layer armor with an increase of areal density of 15.7%. In a word, a 1 mm-thick cover plate provides the best ballistic performance.

It is a pity that the numerical simulation fails to capture the dwell phenomenon which is perhaps attributed to the smaller erosion strain of ceramic adopted in the numerical model. Although it is difficult for one numerical model to accurately predict all aspects of the extremely complex responses in the whole penetration process, this is the direction of our future efforts. Additionally, the results of numerical simulation need to be further verified by a large number of experiments, so as to provide more reliable analysis results and help to design a reasonable and reliable armored structure with a metal cover-layer, which is a research topic for which we need to make great efforts in the future.

## Figures and Tables

**Figure 1 materials-16-03405-f001:**
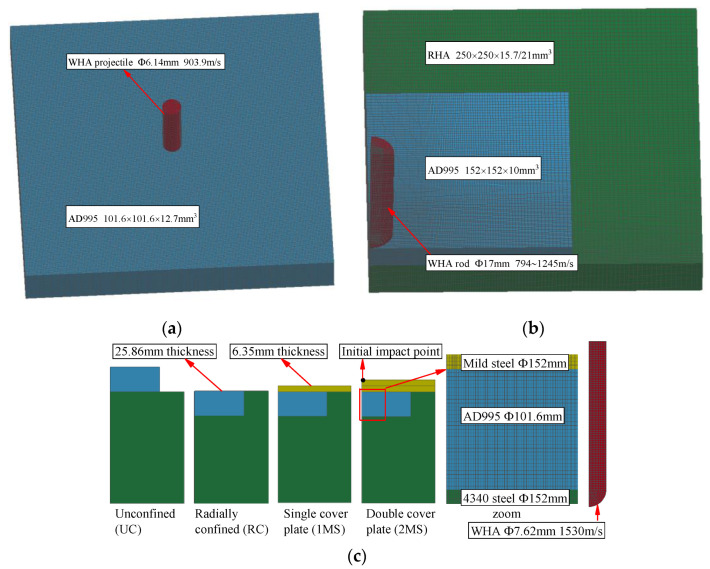
3D modeling of ballistic experiments. (**a**) Full-scale model for test in Ref. [4], (**b**) 1/4 model for test in Ref. [5] and (**c**) 1/4 model for test in Ref. [6].

**Figure 4 materials-16-03405-f004:**
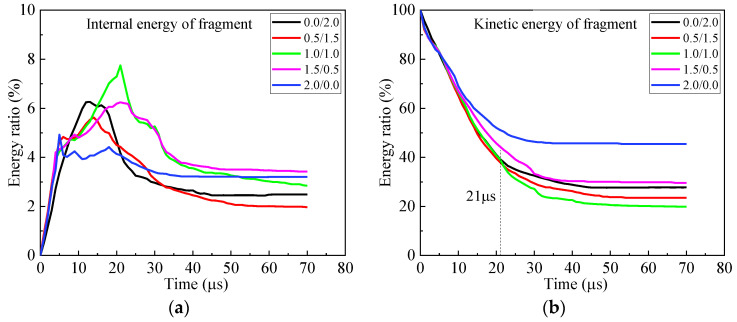
Temporal curves of energy ratios of internal, kinetic and total energies to the initial kinetic energy of the fragment during perforation of armors with constant areal density 4.96 g/cm^2^. (**a**) Internal energy, (**b**) kinetic energy and (**c**) total energy.

**Figure 5 materials-16-03405-f005:**
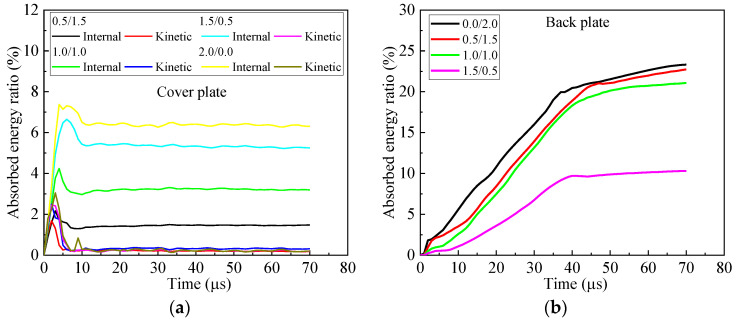
Energy distribution of components in armors with constant areal density 4.96 g/cm^2^. (**a**) Cover plate, (**b**) back plate, (**c**) cover + back plates and (**d**) energy distribution.

**Figure 6 materials-16-03405-f006:**
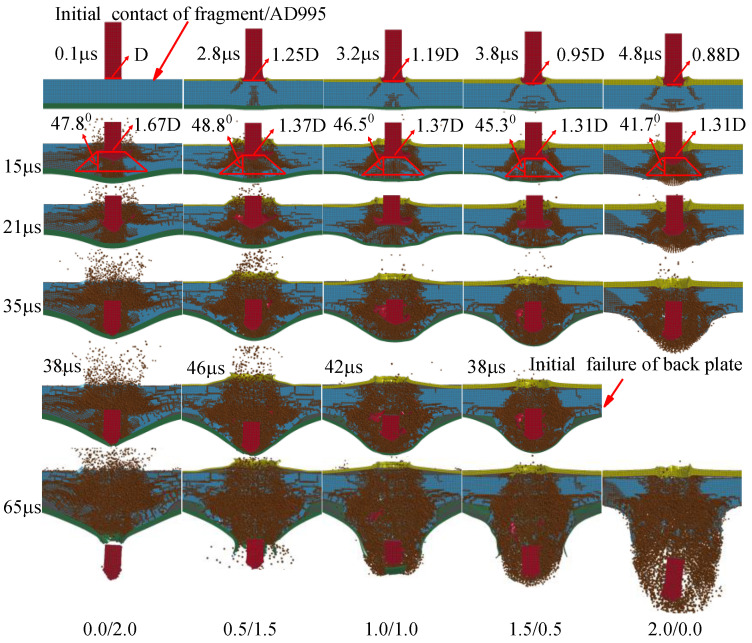
Failure characteristics of fragment and armor with constant areal density 4.96 g/cm^2^.

**Figure 7 materials-16-03405-f007:**
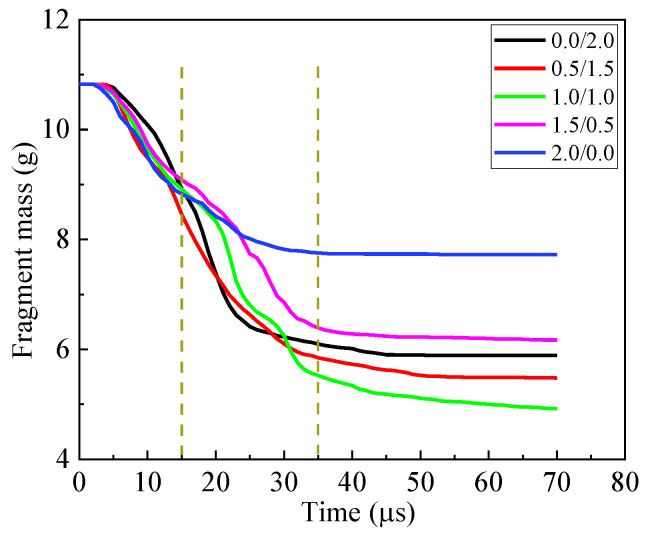
Fragment mass curve for the armor with constant areal density 4.96 g/cm^2^.

**Figure 8 materials-16-03405-f008:**
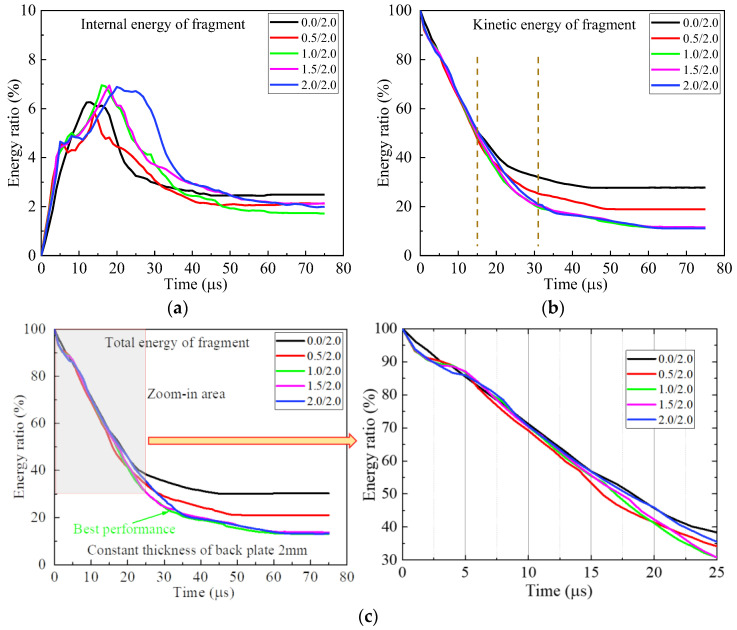
Energy curves of fragment in armors with 2 mm-thick back plate. (**a**) Internal energy, (**b**) kinetic energy and (**c**) total energy.

**Figure 9 materials-16-03405-f009:**
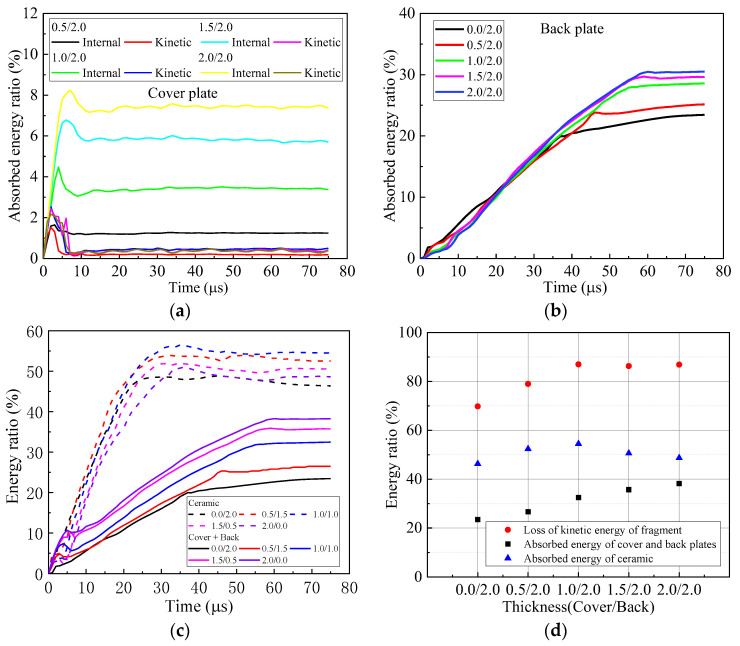
Energy distribution of components in armors with 2 mm-thick back plate. (**a**) Cover plate, (**b**) back plate, (**c**) cover + back plates and (**d**) energy distribution.

**Figure 10 materials-16-03405-f010:**
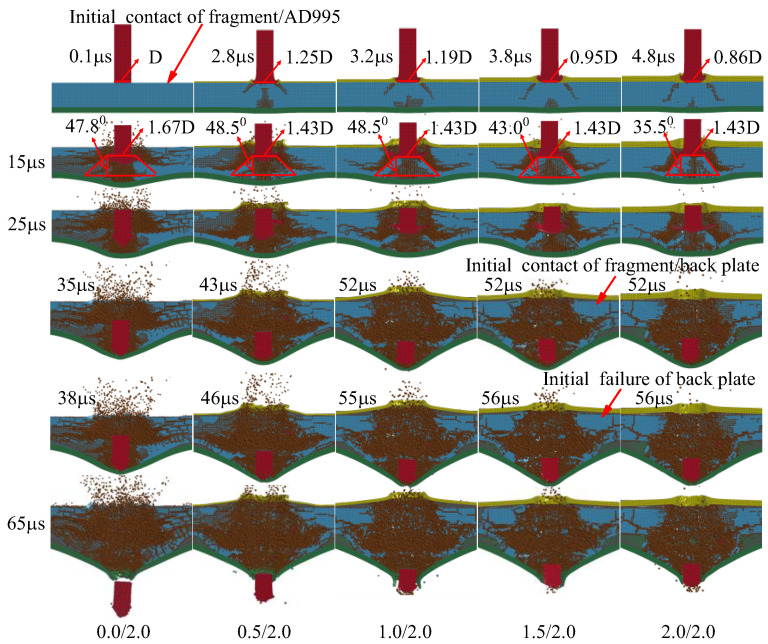
Failure characteristics of fragment and armor with constant 2 mm-thick back plate.

**Figure 11 materials-16-03405-f011:**
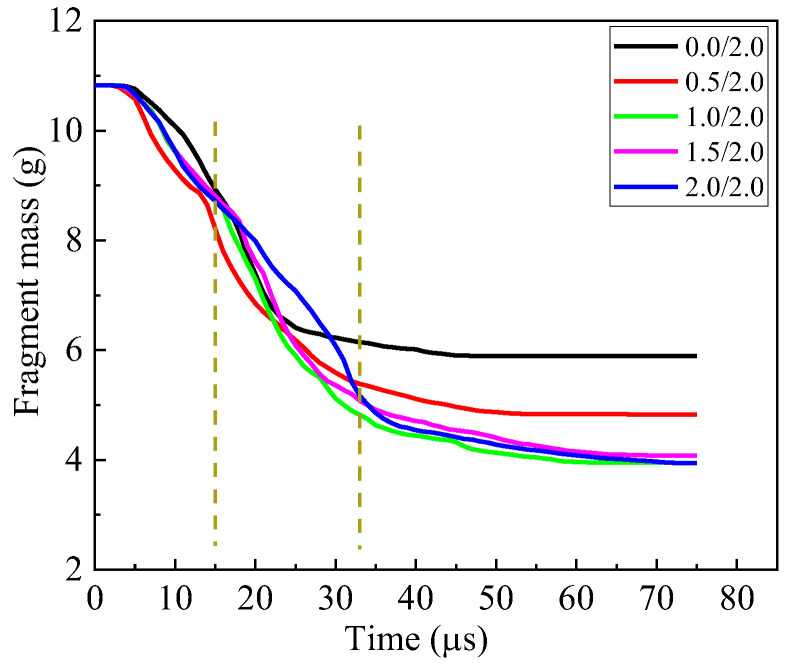
Fragment mass curve for the armor with constant 2 mm-thick back plate.

**Table 1 materials-16-03405-t001:** JH2 model parameters of AD995 alumina [33,34].

Parameter	Symbol	Unit	AD995
Density	*ρ*	kg/m^3^	3890
Shear modulus	*G*	GPa	152.00
Intact strength coefficient	*A*	—	0.93
Intact strength exponent	*N*	—	0.60
Fracture strength coefficient	*B*	—	0.31
Fracture strength exponent	*M*	—	0.60
Strain rate coefficient	*C*	—	0.00
Reference strain rate	${\dot{\varepsilon }}_{0}$	s^−1^	1
Normalized fracture strength	*SFMAX*	—	0.20
Tensile strength limit	*T*	GPa	0.20
Hugoniot elastic limit	*HEL*	GPa	19.00
Pressure at the Hugoniot elastic limit	*P* _HEL_	GPa	1.46
Bulking factor	*β*	—	1
Pressure constant (bulk modulus)	*K* _1_	GPa	220.24
Pressure constant	*K* _2_	GPa	0.00
Pressure constant	*K* _3_	GPa	0.00
Damage constant	*D* _1_	—	0.005
Damage constant	*D* _2_	—	1.000

**Table 2 materials-16-03405-t002:** JC model parameters of WHA [18,33], 4340 steel [24,38], mild steel [17,39] and RHA [19].

Parameter	Symbol	Unit	WHA	4340 Steel	Mild Steel	RHA
Shear modulus	*G*	GPa	160.00	81.80	76.30	80.00
Initial yield stress	*A*	MPa	1506.00	792.00	304.33	1180.00
Hardening coefficient	*B*	MPa	177.00	510.00	422.01	170.00
Hardening exponent	*n*	—	0.120	0.260	0.345	0.280
Thermal softening exponent	*m*	—	1.000	1.030	0.870	1.150
Strain rate coefficient	*C*	—	0.016	0.014	0.016	0.058
Reference strain rate	${\dot{\varepsilon }}_{0}$	s^−1^	1	1	1 × 10^−4^	1
Specific heat	*C* _p_	J/kg.K	134	477	455	440
Melting temperature	*T* _m_	K	1723	1790	1800	1793
Reference temperature	*T* _r_	K	300	300	293	300
Damage constant	*D* _1_	—	2.000	0.050	0.115	0.123
Damage constant	*D* _2_	—	1.770	3.440	1.012	0.000
Damage constant	*D* _3_	—	−3.400	−2.120	−1.768	0.000
Damage constant	*D* _4_	—	0.000	0.002	−0.053	0.694
Damage constant	*D* _5_	—	0.000	0.610	0.526	0.501
Density	*ρ*	kg/m^3^	17,600	7860	7850	7800
Elastic wave velocity	*c*	m/s	4029	4569	4569	4610
*v*_s_-*v*_p_ curve slope	*S* _1_	—	1.237	1.490	1.490	1.730
Grüneisen coefficient	*γ*	—	1.540	2.170	2.170	1.670

**Table 3 materials-16-03405-t003:** Calibration and validation tests.

Refs.	Armors	Dimensions (mm)	Fragments	Dimensions (mm)	Initial Velocity (m/s)
[4]	AD995 ceramic tile	101.6 × 101.6 × 12.7	WHA cylinder	6.14 × 20.86	903.9
[5]	AD995/RHA composite armor	AD995: 152 × 152 × 10 RHA: 250 × 250 × 15.7/21	WHA cylinder with semi-sphere nose	17 × 55	794, 1237, 1245
[6]	(Mild steel cover plate) AD995/4340 steel surround	AD995: 101.6 × 25.86 4340 steel: 152 Mild steel: thickness 6.35	WHA cylinder with semi-sphere nose	7.62 × 76.2	1530

**Table 4 materials-16-03405-t004:** Comparisons of experimental [4] and numerical results.

Evaluation Indicators	Experiment	Simulation	Error (%)
Residual velocity (m/s)	682	690	1.2
Residual mass (g)	6.42	6.08	5.3

**Table 7 materials-16-03405-t007:** Configuration of composite armors based on Heterington equation.

Type	Thickness (mm)		Areal Density of Bi-Layer Armor (g/cm^2^)	$T_{\mathrm{cover}}$ (mm)
	Ceramic	Back Plate		
I	6.0	1.1	3.20	2.2
II	6.5	1.2	3.47	1.9
III	7.0	1.3	3.74	1.6
IV	7.5	1.3	3.94	1.3
V	8.0	1.4	4.21	1.0
VI	8.7	1.5	4.60	0.5
VII	9.5	1.6	4.93	0.0

**Table 8 materials-16-03405-t008:** Configuration and ballistic performance of armor with constant areal density 4.96 g/cm^2^.

Configuration	Cover Plate		Ceramic		Back Plate		Fragment		
	Material	Thickness (mm)	Material	Thickness (mm)	Material	Thickness (mm)	*V*_r_ (m/s)	*m*_r_ (g)	*E*_loss_ (%)
Reference	-	-	AD995	12.7	-	-	690	6.08	64.6
0.0/2.0	-	-	AD995	8.7	RHA	2.0	645	5.89	69.8
0.5/1.5	RHA	0.5	AD995	8.7	RHA	1.5	615	5.48	74.5
1.0/1.0	RHA	1.0	AD995	8.7	RHA	1.0	589	4.92	77.3
1.5/0.5	RHA	1.5	AD995	8.7	RHA	0.5	646	6.17	66.9
2.0/0.0	RHA	2.0	AD995	8.7	-	-	722	7.30	51.2

**Table 9 materials-16-03405-t009:** Configuration and ballistic performance of armor with constant 2 mm-thick back plate.

Configuration	Cover plate (RHA)	Ceramic (AD995)	Back Plate (RHA)	Areal Density		Fragment		
	Thickness (mm)	Thickness (mm)	Thickness (mm)	Value (g/cm^2^)	Increase (%)	*V*_r_ (m/s)	*m*_r_ (g)	*E*_loss_ (%)
Reference	-	12.7	-	4.96	0	690	6.08	64.6
0.0/2.0	-	8.7	2.0	4.96	0	645	5.89	69.8
0.5/2.0	0.5	8.7	2.0	5.35	7.9	587	4.83	79.0
1.0/2.0	1.0	8.7	2.0	5.74	15.7	503	3.94	87.0
1.5/2.0	1.5	8.7	2.0	6.13	23.6	500	4.08	86.3
2.0/2.0	2.0	8.7	2.0	6.52	31.5	497	3.94	86.9

## Data Availability

The data presented in this manuscript are available on request from the corresponding author.

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
