# Peer review of "Numerical Investigation on Protective Mechanism of Metal Cover Plate for Alumina Armor against Impact of Fragment by FE-Converting-SPH Method"

_materials, 2023, doi:10.3390/ma16093405_

Round 1

Reviewer 1 Report

In this study, a numerical investigation were performed on protective mechanism of metal cover plate for alumina armor against impact of fragment by FE-converting-SPH method. Also, the optimization of thickness of RHA cover plate in AD995/RHA composite armor were carried out and several affecting mechanisms involving cover-layer thickness were analyzed and identified.

This study includes some innovations for protective systems exposed to ballistic effects, and this study may also be of interest to the structural engineering community in the related field. For these reasons, I can recommend the publication of this manuscript.

Author Response

We thank the reviewer for your recognition and encouragement of our work.

Reviewer 2 Report

The manuscript has great potential for accepting in this journal. So, it can be accepted in present form.

Author Response

(The authors gave the same response as above.)

Reviewer 3 Report

The manuscript aims to develop a precise numerical model, evaluate the armour performance, and reproduce the ceramic failure by employing the FE-Converting-SPH technique. The applicability and accuracy of the numerical model in terms of algorithm, material model parameters and contact definitions were validated via experiments. The influence of cover-layer thickness on the armour performance was explored based on the results.

The article is written on a good level, and I evaluate it positively.  I recommend publishing it after revisions, focusing on the following comments:

1)    At the end of the introductory part, highlight the novelty and the contribution of Your research.

2)    Give the details of boundary conditions for both

 - the experimental tests (provide documentation/photo/scheme, displaying conditions) and

- numerical analyses - provide a global view of conditions ... at what distance from the armour the initiation speed was defined?; In what position was the armour - horizontal or vertical (with respect to Figures 1 and 5)? What types of elements were selected in the mesh of numeric analyses (size, number or layout), etc.?

3)    Table 5 - Check the "Residual Length of Project (m/s)" - length is not in m/s.

4)    Conclusions should be written more specifically.

5)    Check the units of time on the "x" axes in all the diagrams throughout the manuscript

6)    Similarly, check the units for time in lines 414, 491, ...

7)    There should be a space between the number and the unit. Improve throughout the manuscript.

8)    Unify the writing of References.

Author Response

Comment:

(1) At the end of the introductory part, highlight the novelty and the contribution of your research.

Response:

There are three main innovations in the present manuscript as follows:

(a) FE-converting-SPH method will be adopted to simulate the dynamic response of AD995/RHA and AD995/4340 steel composite armours subjected to the impact of WHA fragment. The numerical model will be validated by various types of test results, focusing not only on the terminal ballistic performance, e.g., DOP in target or residual velocity of fragment, but also on the fracture process and failure pattern of ceramic as well as the residual mass of fragment.

(b) Based on the numerical simulation, the optimization of thickness of RHA cover plate in AD995/RHA composite armour are subsequently carried out. Several affecting mechanisms involving cover-layer thickness are analyzed and identified, such as the pre-damage effect of cover plate upon the ceramic, the shape change of fracture cone of ceramic, the prolonging of interaction time between fragment and ceramic, etc.

(c) The numerical simulation helps to decrease the test number of verified tests and significantly shortens the time and cost of configuration optimization of armours.

The innovations are added at the end of the Section Introduction in the modified manuscript.

(2) Give the details of boundary conditions for both

- the experimental tests (provide documentation/photo/scheme, displaying conditions) and

- numerical analyses - provide a global view of conditions ... at what distance from the armour the initiation speed was defined? In what position was the armour - horizontal or vertical (with respect to Figures 1 and 5)? What types of elements were selected in the mesh of numeric analyses (size, number or layout), etc.?

Response:

The tests in Refs. [3-4] indicate that the boundary of target is free. However, the boundary of target in Refs. [5-6] was not specified. Based on the configuration of test in Refs. [5-6], there is no obvious difference between numerical results when the lateral side of target is free or fixed. The potential mechanism is that the effect of the lateral boundary of target is minor when the lateral size of target is large enough [40]. In this way, the free boundary condition is adopted in the manuscript for tests in Refs. [3-6]. The boundary conditions of tests have been described in the newly revised manuscript. Moreover, the boundary conditions of the numerical analyses in the present manuscript are free unless otherwise specified.

As for ‘at what distance from the armour the initiation speed was defined?’, the velocity when the projectile just hits the target is defined as the initial velocity or impact velocity.

For ‘In what position was the armour - horizontal or vertical (with respect to Figures 1 and 5)?’, the armours in both Figures 1 and 5 (viz. Figures 2 and 6 in the modified manuscript) are placed vertically in tests. Since the influence of gravity is not exerted during the simulation, there is no differences when the target is horizontal or vertical.

For ‘What types of elements were selected in the mesh of numeric analyses (size, number or layout), etc.?’, the 8-node hexahedron element with one Gaussion integral point is adopted to discretize the configuration of test. The element size is depicted in Section 2.3 in the modified manuscript, i.e., uniform 0.5mm for tests in Refs. [3-4] and 0.5mm for the central interaction zone for the tests in Refs. [5-6]. Since the mesh size has influences upon the numerical results, the mesh size we adopted is validated by the test results. In this way, we adopt uniform 0.5mm in the numerical analyses of composite armour with RHA cover plate. The mesh style of the configuration of target and fragment is added and shown in Fig. 1 in the modified manuscript.

(3) Table 5 - Check the "Residual Length of Project (m/s)" - length is not in m/s.

Response:

This is our mistake and it has been corrected in the modified manuscript.

(4) Conclusions should be written more specifically.

Response:

We have re-presented the conclusions of the manuscript following the suggestion of the reviewer. Hopefully, it would be better than the original version.

(5) Check the units of time on the "x" axes in all the diagrams throughout the manuscript

Response:

The units of time on the "x" axes in all the diagrams throughout the manuscript have been checked.

(6) Similarly, check the units for time in lines 414, 491, ...

Response:

The units for time in lines 414, 491, ... have been checked and revised in the modified manuscript.

(7) There should be a space between the number and the unit. Improve throughout the manuscript.

Response:

All of them have been checked and revised in the modified manuscript.

(8) Unify the writing of References.

Response:

The writing of References has been unified according to the Journal template.

Reviewer 4 Report

1. Significant numbers in tables should be adjusted. e.g., 0.6 -> 0.60, 1 -> 1.0.

2. In line 207 -> please leave a space before it.

3. What does "Mild steel" mean? What grade of steel is it? Kindly specify.

4. In table 2. 4340 steel -> 4340 steel.

5. In Line 240-345 -> On what basis do you have such a change and where did such parameters come from?

6. In table 4, kindly correct the position.

7. In Chapter 3, correct the font size.

8. There is no described penetrator, the given velocities do not prove anything with such a dispersion of velocities (Table 3) 794-1245 m/s. What penetrator fired from a gas cannon or rifle ammunition was used? There is no photo of the test stand. Was the velocity measured with Doppler radar or measuring gate? Where was the speed measurement taken? How far from the target?

9. Why are such thicknesses (layers) used for calculations? There is no consideration for this arrangement.

10. In the summary, the authors refer to the correct formulation of the FE-con-SPH method, however, the key check of the conversion itself is missing. Describing a ready-made conversion subroutine without a thorough analysis of individual variables does not allow for a thorough analysis of the phenomenon of the description of the mechanism of ceramic material destruction. Apart from one photo, the authors do not show a comparison of the experiment with numerical analysis.

Author Response

Comment:

(1) Significant numbers in tables should be adjusted. e.g., 0.6 -> 0.60, 1 -> 1.0.

Response:

They have been checked and revised the significant numbers throughout this manuscript.

(2) In line 207 -> please leave a space before it.

Response:

It has been checked and revised in the modified manuscript.

(3) What does "Mild steel" mean? What grade of steel is it? Kindly specify.

Response:

In Ref. [6], it is only vaguely termed as ‘mild steel’, however, the grade, mechanical properties of ‘mild steel’ are unspecified. We tried our best to find other published works by the authors of Ref. [6] to see if there was any concrete description of ‘mild steel’. Unfortunately, we found nothing. Therefore, we have followed this ambiguous appellation. We have used the relevant parameters measured by experiment [38] and widely adopted for numerical research of ‘mild steel’ during ballistic impact events [17].

(4) In table 2. 4340steel -> 4340 steel.

Response:

It has been checked and revised in modified manuscript.

(5) In Line 240-345 -> On what basis do you have such a change and where did such parameters come from?

Response:

The values of erosion parameters for materials are highly empirical and grid-dependent. Their determinations are through repeated numerical attempts, finally getting the parameters that match a series of test results. A similar expression is shown in the newly revised manuscript.

(6) In table 4, kindly correct the position.

Response:

It has been checked and revised in modified manuscript.

(7) In Chapter 3, correct the font size.

Response:

It has been checked and revised in modified manuscript.

(8) There is no described penetrator, the given velocities do not prove anything with such a dispersion of velocities (Table 3) 794-1245 m/s. What penetrator fired from a gas cannon or rifle ammunition was used? There is no photo of the test stand. Was the velocity measured with Doppler radar or measuring gate? Where was the speed measurement taken? How far from the target?

Response:

The penetrator is a WHA fragment, and its size is specifically described in Table 3 of the newly revised manuscript.

For the speed range of 794-1245m/s, we have modified it according to the reviewer's opinion, and specified the specific speeds as 794,1237 and 1245 m/s, which are the velocities adopted to validate the model and are shown in Table 3.

The penetrator was fired by gas gun in Refs. [3-4], smoothbore gun in Ref. [5] and powder gun in Ref. [6], respectively. In this way, the fragment does not have rotation about its axis.

As for ‘There is no photo of the test stand.’, the configuration of target and fragment named as Figure 1(a), (b) and (c) are added in the modified manuscript.

As for ‘Was the velocity measured with Doppler radar or measuring gate? Where was the speed measurement taken? How far from the target?’, there is a lack of detailed description of relevant contents in Refs. [5-6]. On the other hand, the measurement device only affects the accuracy of measurement, which should have minor influence on the ballistic performance of target if they are correctly measured.

(9) Why are such thicknesses (layers) used for calculations? There is no consideration for this arrangement.

Response:

It is based on the consideration of the ceramic thickness and areal density of armour needed in some researches of defending small-caliber threats in Refs. [36,45-46]. Moreover, the selection of the thickness of each layer comes from the optimization result of Heterington equation [2]. In the first three paragraphs of Section 3.1 of the modified manuscript, the specific reasons for choosing such ceramic thickness and areal density of armour are given.

(10) In the summary, the authors refer to the correct formulation of the FE-con-SPH method, however, the key check of the conversion itself is missing. Describing a ready-made conversion subroutine without a thorough analysis of individual variables does not allow for a thorough analysis of the phenomenon of the description of the mechanism of ceramic material destruction. Apart from one photo, the authors do not show a comparison of the experiment with numerical analysis.

Response:

The numerical model including the material properties, the element size, the contact algorithm as well as the conversion criterion is lumped together to be validated by the test results in open references. However, the interaction time of target and fragment is usually within 1 ms and there would be dust cloud and sparkles around the impact point of target. Non-destructive, perspective and high-frequency test equipment should be adopted to obtain the intermediate process of penetration, e.g., x-ray image. Due to the characteristics of the impact test, there are usually only terminal ballistic results, such as DOP in target, residual velocity of fragment, residual mass of fragment, etc., for example in Refs. [5-6]. It is cost or even impossible to record the penetration process of the armor in test. This is one main reason we carried out the numerical simulation to reveal the interaction process of target and fragment. Fortunately, a series of x-ray images of penetration process of armor are reported in Refs. [3-4], which could provide a reliable experimental basis to verify our numerical model.

In order to verify the numerical model reliably, we tried our best to validate the numerical model by a variety of tests with different configurations of target and fragment. The thickness of target is from thin to semi-infinite. The shape of fragment is blunt or hemi-spherical. The impact velocity falls in a broad range. The DOP in target and residual velocity of fragment as well as the morphologies of target and fragment after shot are compared between test and numerical simulation. It is glad to obtain a good agreement between each other. In this way, the numerical model is verified reliably.

Moreover, with the improvement of measuring technique, more intermediate processes would be recorded during test. The numerical model would be further validated by the upcoming test results in the future.

Reviewer 5 Report

The paper

“Numerical investigation on protective mechanism of metal cover plate for alumina armor against impact of fragment by FE-converting-SPH method”,

By Linlong Dou et al.,

Discusses an accurate numerical model to simulate and predict the ballistic response of ceramic armor subjected to high-velocity impacts (HVI).

The paper is quite interesting and well in line with the scope of Materials MDPI.

Nevertheless, some conceptual and editorial concerns should be addressed before granting full acceptance. These are enlisted here below.

1.      There are some slight issues with the font size and type in several sections. Please uniform it everywhere according to the guidelines for authors.

2.      Also, double-check throughout the whole text for typos and grammar mistakes

3.      Table 1, Table 2, and elsewhere: please use the same number of decimal digits for similar quantities. E.g. D1 and D2 are respectively listed as 0.005 and 1 (with no decimal digits). In Table 2, the same quantities range from five to none decimal units.

4.      The strain rate coefficient is reported as 0.0 in Table 1. Is this a typo?

5.      The main weakness of this otherwise interesting research work is that it is essentially purely numerical. Apart from the test image in Figure 2, are there any experimental data to eventually support these results?

Author Response

Comment:

(1) There are some slight issues with the font size and type in several sections. Please uniform it everywhere according to the guidelines for authors.

Response:

They have been checked and revised throughout this manuscript according to the guidelines for authors.

(2) Also, double-check throughout the whole text for typos and grammar mistakes

Response:

They have been checked and revised, for example:

  • The ‘through the use of the’ in the line 128 of the modified manuscript is changed to ‘by’;
  • The ‘tests’ in the line 205 of the modified manuscript is changed to ‘experiments’;
  • The ‘would’ in the line 435 of the modified manuscript is changed to ‘will’;
  • The ‘the’ in the line 482 of the modified manuscript is added in front of ‘best’;
  • The ‘ms’ in the line 559 of the modified manuscript is changed to ‘μs’, etc.

Limited by the space, we do not list all modifications in the reply.

(3) Table 1, Table 2, and elsewhere: please use the same number of decimal digits for similar quantities. E.g. D1 and D2 are respectively listed as 0.005 and 1 (with no decimal digits). In Table 2, the same quantities range from five to none decimal units.

Response:

They have been checked and revised in modified manuscript.

(4) The strain rate coefficient is reported as 0.0 in Table 1. Is this a typo?

Response:

As reported in Ref. [Ji M, Li H, Zheng J, et al. An experimental study on the strain-rate-dependent compressive and tensile response of an alumina ceramic[J]. Ceramics International. 2022, 48(19): 28121-28134.], the alumina ceramic exhibits minor strain rate sensitivity before failure in the range of strain rate between 10-4~103 /s.

Also, in Ref. [Shafiq M, Subhash G. Which one has more influence on fracture strength of ceramics: pressure or strain rate? Cham, Switzerland. Conference proceedings of the society for experimental mechanics series, 2017]. It was pointed out more clearly that the strain rate coefficient (C) was extremely small (0.003-0.012) and more than an order of magnitude smaller compared to the strength coefficient (A) and strength exponent (N) for the JH2 model used in our simulation. In this way, we ignored the strain rate effect of ceramic in numerical analyses for approximation. The references [22, 33-34] adopted the same approximation.

(5) The main weakness of this otherwise interesting research work is that it is essentially purely numerical. Apart from the test image in Figure 2, are there any experimental data to eventually support these results?

Response:

This is a very helpful suggestion. It is indeed the short slab of our work that we have not carried out any tests. Firstly, we adopted a series of test results in open references to validate the numerical model. The intermediate process and terminal parameters of target and fragment are compared and a good agreement is obtained, as shown in Figs. 2-3 and Tables 4-6 in the modified manuscript. It confirms the reliability of the numerical model. Secondly, as depicted in the end of Introduction, the numerical simulation helps to decrease the test number of verified tests and significantly shortens the time and cost of configuration optimization of armours. Based on the numerical analyses, we would carry out verified tests in the future. However, limited by the cost and long-term of test designation, we would like to share our numerical analyses and exchange the academic opinions with the peers in the first place. Finally, the research results on the optimal thickness of metal cover-layer are also consistent with the suggestions in Ref. [46], and are explained at the end of Section 3.

Round 2

Reviewer 3 Report

I have carefully studied the improved version of the manuscript and consider it publishable in its current form.

Author Response

(The authors gave the same response as above.)

Reviewer 4 Report

Thank you for the changes made.

Please only correct minor typos such as:

1. line 340, 341.

2. "Reference strain rate" symbol in Tables 2 and Tables 1

Author Response

Comment:

Please only correct minor typos such as:

  1. line 340, 341.
  2. "Reference strain rate" symbol in Tables 2 and Tables 1

Response:

They have been checked and revised in the latest revised manuscript.

Reviewer 5 Report

The authors of the paper provided an exhaustive reply to this reviewer. Hence, having all the major remarks been addressed, this Reviewer suggests the acceptance of the submitted paper and its publication, after careful grammar checking and proofreading, and after having addressed the following minor (editorial) remaining issues:

- In tables 1 and 2, there is an issue at line 9 (‘Reference strain rate’), column 2 (‘Symbols’).

- On page 10 line 341, there is an issue with the cross-reference to an equation (probably Eq. 2?)

- figures 5.d, 9.a, and 9.d: it would be better to have a single legend for the whole subplot

- in the current form, the Conclusions are a bit lengthy and can use some shortening.

Author Response

Comment:

(1) In tables 1 and 2, there is an issue at line 9 (‘Reference strain rate’), column 2 (‘Symbols’)

Response:

They have been checked and revised in the latest revised manuscript.

(2) On page 10 line 341, there is an issue with the cross-reference to an equation (probably Eq. 2?)

Response:

They have been checked and revised in the latest revised manuscript.

(3) Figures 5.d, 9.a, and 9.d: it would be better to have a single legend for the whole subplot

Response:

This is a very helpful suggestion. According to the reviewer's opinion, we have revised all the pictures with this requirement in the latest revised manuscript.

(4) In the current form, the Conclusions are a bit lengthy and can use some shortening

Response:

According to the reviewer's opinion, under the premise of ensuring the accurate description of the research conclusions, we have shortened the statement of the conclusion as much as possible. Deleted some contexts that perhaps have repeated descriptions or can be identified without description, such as' armor with constant real density of 4.96g/cm2' in line 631, ‘The confinement is better when the 4.96 g/cm2-areal density AD995/RHA armor covered by RHA cover plate with thickness more than 1.0mm.’ in lines 639-640 and ‘in ceramic' in line 641, etc, in the latest revised manuscript.